# bHLH010/089 Transcription Factors Control Pollen Wall Development via Specific Transcriptional and Metabolic Networks in *Arabidopsis thaliana*

**DOI:** 10.3390/ijms231911683

**Published:** 2022-10-02

**Authors:** Zesen Lai, Jianzheng Wang, Shi-Qing Peng, Fang Chang

**Affiliations:** 1School of Life Sciences, Hainan University, Haikou 570228, China; 2State Key Laboratory of Genetic Engineering, Ministry of Education Key Laboratory of Biodiversity Sciences and Ecological Engineering and Institute of Biodiversity Sciences, Institute of Plant Biology, School of Life Sciences, Fudan University, Shanghai 200438, China; 3Key Laboratory of Biology and Genetic Resources of Tropical Crops, Ministry of Agriculture, Institute of Tropical Bioscience and Biotechnology, Chinese Academy of Tropical Agricultural Sciences, Haikou 571101, China

**Keywords:** anther, pollen wall development, *bHLH010*, *CSLB03*, *Arabidopsis thaliana*

## Abstract

The pollen wall is a specialized extracellular cell wall that protects male gametophytes from various environmental stresses and facilitates pollination. Here, we reported that bHLH010 and bHLH089 together are required for the development of the pollen wall by regulating their specific downstream transcriptional and metabolic networks. Both the exine and intine structures of *bhlh010 bhlh089* pollen grains were severely defective. Further untargeted metabolomic and transcriptomic analyses revealed that the accumulation of pollen wall morphogenesis-related metabolites, including polysaccharides, glyceryl derivatives, and flavonols, were significantly changed, and the expression of such metabolic enzyme-encoding genes and transporter-encoding genes related to pollen wall morphogenesis was downregulated in *bhlh010 bhlh089* mutants. Among these downstream target genes, *CSLB03* is a novel target with no biological function being reported yet. We found that bHLH010 interacted with the two E-box sequences at the promoter of *CSLB03* and directly activated the expression of *CSLB03.* The cslb03 mutant alleles showed *bhlh010 bhlh089*–like pollen developmental defects, with most of the pollen grains exhibiting defective pollen wall structures.

## 1. Introduction

The pollen wall is a specialized multiple-layer outer surface of a pollen and is essential for plant reproduction. Its primary role is to provide structural and physical support for the gametophytes and protection from various environmental damage as well as function in male–female interaction and pollination [1]. Despite the morphological diversity of pollen wall ontogeny, the fundamental structure of the pollen wall is generally conserved among taxa, and is commonly composed of an outer layer, the exine, and an inner layer, the intine [2,3,4]. These two layers of the pollen wall are made up of distinct components, with the exine consisting of sporopollenin that is derived from the tapetum and the intine mostly consisting of hydrolytic enzymes, cellulose, and pectic polymers that are constructed from the microspores themselves [5,6,7].

The development of the pollen primexine in higher plants initiates at the tetrad stage, when the microsporocytes finish meiosis, the microspores (MSps) are entirely covered by the callose wall, and the primary exine is formed around the distinct microspores. Along with the sporopollenin precursors, those from the tapetum layer, being deposited onto the primexine, the primexine structure that contains the basis of bacula and tectum is built up. After the dissolution of the callose wall and the release of microspores, the primexine structure increases in thickness and size with the polymerization of sporopollenin. The mature exine structure is completed by the bicellular pollen stage [3,8]. Previous studies demonstrated multiple genes that are involved in regulating the formation of the primary pollen wall, including *defective in exine formation 1* (*DEX1*)*, No Primexine And Plasma Membrane Undulation* (*NPU*)*, ruptured pollen grain 1* (*RPG1*)*, exine formation defect* (*EFD*)*, hackly microspore* (*HKM*)*, no exine formation 1* (*NEF1*), *and humidity-sensitive genic male sterility 1* (*HMS1*) [5,9,10,11,12,13,14,15,16,17,18,19]. The synthesis of sporopollenin and its precursors requires various regulatory factors and proteases, for instance, acyl-CoA synthetase 5 (ACOS5), fatty acyl-CoA reductase (FAR), eceriferum3 (CER3), wax deletion A1 (WDA1), cytochrome P450 703A2 (CYP703A2), polyketide synthases A (PKSA), polyketide synthases B (PKSB), Male Sterile 188 (MS188), MYB transcription factor 103 (MYB103), and tetraketide A-pyrone reductase 1 (TKPR1) [20,21,22,23,24,25,26,27,28,29,30,31,32]. In addition, transporters including ABCG transporter 11 (ABCG11), OsABCG15, AtABCG22, AtABCG26, and AtABCG31 are also needed to transport the sporopollenin precursors from the tapetal cells onto the pollen wall [33,34,35,36,37,38,39,40,41,42].

In comparison, there are few studies on the development of pollen intine [4]. Genes that are related to the formation of the pollen intine development include *cellulose synthase protein A1* (*CESA1*)*, CESA3, UDP-sugar pyrophosphorylase* (*USP*)*, glycoside hydrolase family* (*GH*)*, carbohydrate esterase family* (*CE*)*, glycosyl transferase family* (*GT*)*, cell wall invertase 2* (*CWI2*), and *fasciclin-Like arabinogalactan protein 3 (FLA3)* [43,44,45,46,47,48,49,50,51]. Moreover, metabolite transporters include ABCG transporter 1 (ABCG1), ABCG16, and reversibly glycosylated polypeptides 1/2 (RGP1/2), which are also involved in the formation of pollen intine [52]. However, the functions and upstream regulatory pathways of most of metabolites of pollen are still not clear.

As the tapetum acts as a supplier of sporopollenin precursors for pollen exine development, the normal development and programmed cell death (PCD) of the tapetum are essential for pollen development, including pollen exine formation [5]. Either abnormal development or advanced/delayed PCD of a tapetum leads to impaired pollen wall development and reduced male fertility [53,54,55,56,57,58,59,60,61,62,63,64,65]. A genetic transcriptional pathway has been demonstrated to function in regulating the development and PCD of a tapetum, from the upstream to the downstream, including the following transcription factor (TF) genes: *dysfunctional tapetum 1* (*DYT1*), *bHLH010/bHLH089/bHLH091*, *defective in tapetal development and function 1* (*TDF1*)/*MYB35*, and *aborted microspores* (*AMS*), *MYB103* [66,67,68,69,70,71,72,73,74]. Although all of the above TF genes were confirmed to be required for tapetum development, whether they function in pollen wall formation and the possible regulatory pathways remains not very clear. Take *bHLH010* and *bHLH089* as examples; although these two genes were demonstrated to be redundantly required for male fertility at both normal and evaluated temperatures, by forming both feedforward and positive feedback regulatory loops with DYT1, a gate-keeper bHLH transcription factor of the anther tapetal transcriptional networks [71,72,75,76], their function in pollen wall development, especially at the metabolomic and transcriptomic level, remains elusive.

In this study, with pieces of evidence from cytological, metabolomic, and transcriptomic levels, we demonstrated that the bHLH010 and bHLH089 TFs together play significant roles in regulating the development of both pollen exine and intine, by regulating the expression of metabolic enzyme-encoding and transporter-encoding genes, as well as affecting the accumulation of polysaccharides, cellulose, flavonols, and other metabolites that are required for the formation of pollen walls. Moreover, a number of genes with unknown functions were found to be involved in fertility, which provided a certain basis for subsequent studies on male fertility. 

## 2. Results

### 2.1. bhlh010 bhlh089 Pollen Exhibited Collapsed Structure and Severely Reduced Germination Rate

As previous studies have revealed, bHLH010 and bhLH089 together are required for the anther development [71,72,75,76]. Neither a *bhlh010* nor *bhlh089* single mutant showed obvious fertility defects, whereas the *bhlh010 bhlh089* double mutants showed defective anther development, with obviously reduced anther size and pollen amount [75]. In addition, we found that the silique length of *bhlh010 bhlh089* plants was obviously decreased (Appendix A). Here in this study, we questioned whether they are required for the development of pollen exine and/or intine. First, the stage 13 anthers were observed under scanning electron microscopy (SEM). The WT anthers dehisced normally and produced a large amount of pollen grains (Figure 1A,D). In contrast, anthers of early *bhlh010 bhlh089* flowers totally failed to dehisce (Figure 1B). Although anthers of late *bhlh010 bhlh089* flowers could dehisce, they produced less pollen, and all the pollen grains were collapsed (Figure 1C,E). In contrast, the percentage of collapsed pollen was about 1.9% in WT (Figure 1L).

In addition, the WT pollen had a clear and completely well-organized networklike reticulate exine surface (Figure 1F,G), whereas all the *bhlh010 bhlh089* pollen exhibited severely destructive exine, with no reticulate structure being observed (Figure 1H,I). To further test the viability of these pollen grains, the pollen germination rates of the WT and *bhlh010 bhlh089* double mutant were analyzed (Figure 1J,K). The pollen of wild type and *bhlh010 bhlh089* was spread on the germination medium and cultured in an incubator (22 °C, 6 h). The germination was observed and counted whether there was pollen tube protruding (because the pollen germination rate of double mutant plants was low within 2 h, the pollen germination was only counted at the 5 h time point to ensure the accuracy of the results). In comparison with the WT pollen that showed an 87.8% germination rate, only a small portion (13.2%) of *bhlh010 bhlh089* pollen successfully germinated (Figure 1M). 

### 2.2. Both the Exine and Intine Developments Were Abnormal in bhlh010 bhlh089

Then the development of the pollen wall was observed. First, results from staining analyses (orange red for exine, and blue for intine) showed that the thicknesses of both the exine and the intine of the *bhlh010 bhlh089* pollen were uneven and much thinner than that of the WT pollen (Figure 2A–D). The average thickness of the WT pollen exine was 1.20 μm, whereas that of *bhlh010 bhlh089* was severely reduced to 0.36 μm, and that of the *bhlh010 bhlh089* intine was also dramatically reduced, from 0.34 μm for the WT to 0.16 μm for the double mutant (Figure 2C,D).

To verify the stages when the pollen exine or intine defects started, the semi-thin sections of *bhlh010 bhlh089* and WT anthers were performed to the pollen wall staining analyses. In WT pollen grains, the exine signal was faintly detected at anther stage 8 and significantly increased at level during anther stage 9–12 [77,78], suggesting the formation and development of exine; the intine signal started to appear at anther stage 10, later than the arising of the exine signal. However, in the *bhlh010 bhlh089* pollen, although the emergence and developmental stages of the pollen exine were normal, the signal of the pollen intine was obviously weaker (Figure 2E, Appendix A). In particular, only 36.3% of pollen that showed a detectable intine signal at anther stage 10 went through the developmental processes, although their final pollen wall architectures were distorted; the other 63.7% of the *bhlh010 bhlh089* pollen that failed to form the intine subsequently underwent degeneration (Figure 2E). These above results strongly supported that bHLH010 and bHLH089 play important roles in regulating the formation of both the pollen exine and intine.

### 2.3. Metabolomic Comparison of WT and bhlh010 bhlh089 Inflorescences

To explore the downstream metabolites and anabolism pathways that are regulated by bHLH010/bHLH089 and related to pollen wall development, untargeted metabolomics was performed with WT and *bhlh010 bhlh089* inflorescences. We collected six inflorescence tissues from WT and *bhlh010 bhlh089* as biological replicates. In total, 494 known metabolites were detected (Appendix A). As shown in the principal component analysis (PCA) score plots, all quality control (QC) samples clustered well, and the dispersion of QC samples was obviously lower than that of samples to be detected (Figure 3A), indicating that the system stability was adequate. These known metabolites belonged to 48 classes, with majority of these metabolites belonging to glycerophospholipids, flavonoids, fatty acyls, prenol lipids, carboxylic acids and derivatives, and organooxygen compounds. To screen differentially accumulated metabolites between Group W1 (WT) and Group D1 (*bhlh010 bhlh089* double mutant), these two groups were analyzed by a multivariate PCA method and by orthogonal projections to latent structure discriminant analysis (OPLS-DA) (Figure 3B). There was an obvious separation trend between Group W1 and Group D1. In addition, the OPLS-DA models were validated by permutation tests to assess whether the models were overfitting. The results of the permutation test suggested that the models were not overfittings (Figure 3C). The S-plots of Group W1 and Group D1 were also constructed (Figure 3D). The above results indicated that there were significant differences between the WT and the *bhlh010 bhlh089* metabolomes.

Therefore, a comparative analysis was performed to identify the differentially accumulated metabolites (DAMs) between the WT and the *bhlh010 bhlh089* metabolomes. A total of 273 DAMs (fold change |FC| > 1.5) were screened out, including 224 DAMs that were downregulated and 49 that were upregulated in the *bhlh010 bhlh089* mutant (Appendix A). The cluster analysis showed that these 273 DAMs were highly enriched in metabolites of fatty acyls, flavonoids, and carbohydrate derivative families (Figure 3E); and metabolites that showed a significant decrease (more than 1.5-fold) in *bhlh010 bhlh089* were enriched in metabolic pathways that involved the phenylalanine biosynthesis, anabolism of starch and sugar, fructose and mannose metabolism, galactose, flavonol, and fatty acid derivatives (Figure 3F). Through the analyses of the top 10 significantly increased and decreased metabolites in a *bhlh010 bhlh089* mutant (D1), the phospholipid and flavonol metabolic pathways were found to be significantly downregulated (Figure 3G), suggesting that bHLH010 and bHLH089 likely positively regulate these two pathways. Therefore, through the comparison metabolic analyses, we obtained the metabolic pathways that were likely regulated by bHLH010 and bHLH089.

### 2.4. bHLH010/089 Regulate the Expression of Genes Involved in Metabolic Pathways

To find out the bHLH010/bHLH089 downstream genes that were involved in the above metabolic pathways, the original transcriptomic raw data of anthers of the WT and *bhlh010 bhlh089* double mutants [79] were reanalyzed. After the reanalysis, the expressions of 3207 genes were found to be obviously changed (Appendix A). Among them, 722 were obviously upregulated (log[fold change (FC)] ≥ 1 and *p* < 0.01) and 2485 were obviously downregulated in *bhlh010 bhlh089* anthers. The 722 upregulated genes were highly enriched in the functional aspects of the sulfur compound metabolic process, fatty acid metabolic process, and secondary metabolic process (Figure 4A) in *bhlh010 bhlh089* anthers, whereas the downregulated genes were enriched in the functional aspects related to the anther development, pollen exine formation, pollen wall assembly, pollen development, lipid localization, regulation of signal transduction, regulation of cell communication, and secondary metabolic process (Figure 4B). To further explore the correlation between the anther transcriptome and metabolites of *bhlh010 bhlh089* double mutants, transcriptomic and metabolomic data were combined to analyze the pathways associated with their common downregulation pathways. The results showed that pathways involving MAPK signaling, phytohormone metabolism, starch and sugar metabolism, fatty acid biosynthesis, flavonols, and phenylalanine derivatives were highly correlated between the transcriptomic and metabonomic data (Figure 4C).

Among these metabolic-pathway-related genes, a portion of them have been reported to be involved in pollen wall formation (Figure 4D–G). For example, the *CSLs* family genes have been reported to be involved in cellulose synthesis [80], the *LAP6* gene is known as an important regulator for pollen exine development [32,81], and some genes of ABC transporter family are involved in metabolite transport in pollen wall formation [33,36]. These results suggested that bHLH010/089 contributed to the development of the pollen exine and intine by regulating both the metabolite synthesis and transports. In addition, in the expression of 15 of the UDP-glucosyltransferase (UGT) family genes, some UGT genes are involved in sugar metabolic pathways and may play an important role in stress or defense responses [82,83]. There were also obvious changes in *bhlh010 bhlh089* anthers (Figure 4G). However, whether they are directly regulated by bHLH010 and/or bHLH089 remains unknown. 

### 2.5. Screening of Genes Activated by bHLH010 or/and bHLH089 

We then performed quantitative real-time PCR (qRT-PCR) analysis and transcriptomic activation activity assay to confirm the expression differences of candidate genes in the WT and *bhlh010 bhlh089* mutants and verify the direct regulation of bHLH010 or/and bHLH089 to these genes, respectively. First, the expression changes of the following genes were confirmed to be strongly downregulated in *bhlh010 bhlh089* anthers (Figure 5A,B): the *CWIN4* gene that regulates nectar production in floral organ [84,85]; the *CSLD1* and *CSLD4* genes that are essential for cellulose deposition and normal pollen tube growth [80]; the *CSLD5* gene that plays a crucial role in root hair development [86,87]; the *UGT73B1*, *UGT73B3*, *UGT76C4*, *UGT76E2*, *UGT91B1*, and *IAGLU* genes that may be involved in sugar metabolic pathways and stress responses [82,83,88,89,90]. Although the functions of a portion of the above genes have been reported to regulate cell wall formation and/or stress responses in various tissues, their functions in the regulation of pollen development, especially in the regulation of pollen wall formation, remain unknown yet.

In addition, *SUS3*, *PLIP2*, *LAP6*, *AT1G06990*, *TPS6*, *ACOS5*, and *AT1G02813* were significantly downregulated in the anther of *bhlh010 bhlh089* (Figure 5C). SUS3 is known to play an important role in sucrose metabolism (Angeles-Nunez and Tiessen, 2010; Angeles-Nunez and Tiessen, 2012), PLIP2 is a glycerolipid A1 lipase and is regulated by hormone signaling [91], *AT1G06990* encodes a GDSL motif–containing lipase [92], TPS6 is highly expressed in inflorescence and participates in the synthesis of trehalose [93], and *ACOS5* and *LAP6* are key genes regulating the development of the pollen exine [31,32,81]. The ABC transporter family is likely to be involved in the process of small-molecule metabolite transport and the formation of the pollen exine and intine. *ABCG1*, *ABCG16*, *ABCG26*, and *ABCG31* that have been reported to be involved in the development of pollen exine [37,39,42] were confirmed to be significantly downregulated in *bhlh010 bhlh089* anthers (Figure 5D).

### 2.6. The CSLB03 Gene Is a Direct Downstream Target of bHLH010

To further verify the possibility that some of the above enzymes or transporter genes were directly activated by bHLH010 or/and bHLH089, we performed dual-luciferase transcriptomic activation activity assay. Among the detected candidate genes, the expressions of *CSLD5*, *CSLD6*, *UGT85A5*, *SUS3*, *AT1G02813*, and *CSLB03* were found to be directly activated by bHLH010 (Figure 6A); and the expressions of *CSLD5*, *CSLD6*, *FRA8*, *UGT85A5*, *LAP6*, *ABCG9*, and *TSM**1* (*tapetum-specific methyltransferase 1*) were found to be activated by bHLH089 (Figure 6B). The expressions of *CSLD5*, *CSLD6*, *UGT85A5*, and *LAP6* were found to be activated by either bHLH010 or bHLH089, although bHLH010 had low activation of *UGT85A5* and *LAP6* (Figure 6C). In contrast, the expressions of *CSLD4*, *MGT5*, and *ABCG17* were not activated by either bHLH010 or bHLH089 (Appendix A), suggesting that bHLH010/089 may indirectly regulate the expression of these genes at the transcriptional level.

As bHLH TFs usually activate the expression of their targets by binding to the E-box motif at the promoter, we synthesized two fragments (P1, P2) representing the two DNA sequences at the *CSLB03* promoter containing the E-box motif (Figure 6D). The electrophoretic mobility shift assay (EMSA) results showed that bHLH010-HIS could bind to both the P1 and P2 fragments. The band signal of the bHLH010-HIS protein and P1 probe complex was significantly reduced after the addition of the P1 competitor probe (50×), and that of the bHLH010-HIS protein and P2 probe complex was significantly reduced by adding the P2 competitor probe (100×), suggesting that the bHLH010-HIS protein specifically interacted with both the P1 and P2 E-boxes of the *CSLB03* promoter (Figure 6E,F). These above results together revealed that *CSLB03* is a novel direct target of bHLH010 in the regulation of pollen wall morphogenesis.

### 2.7. CSLB03 Gene Is Required for Pollen Wall Development

To uncover the function of *CSLB03* in pollen development, we first generated the *proCSLB03:GUS* transgenic plants to verify the expression of *CSLB03* in anther. The results showed that *CSLB03* is expressed in both tapetum cells and microspores (Appendix A), which is similar to *bHLH010*. Semi-thin section results showed that *CSLB03* was expressed in microspores at anther stage 9 and afterward, which was the period of pollen wall development. This expression pattern of *CSLB03* suggested that it is likely to play a role in pollen.

Then, we observed the pollen developmental phenotypes of *cslb03* T-DNA insertion knock-out alleles (*cslb03-1* and *cslb03-2*). Both plants showed no obvious vegetative defects, but exhibited severely reduced fertility, including a shorter silique length, a slightly smaller anther size, and severely reduced pollen amounts in each anther (Figure 7A–D,P,Q). Compared with the siliques of WT (1.429 ± 0.04059 cm), those of the *cslb03-1* and *cslb03-2* mutants were obviously shorter (0.6712 ± 0.05277 and 0.6561 ± 0.04082 cm). In addition, similar to that observed in the *bhlh010 bhlh089* double mutants (Figure 1H,J,K), pollen grains of both *cslb03* mutants were distorted and malformed (Figure 7I–K,Q), and the germination rate of the *cslb03* pollen was severely reduced (Appendix A). Statistical analysis showed that over 63.4% of the *cslb03* pollen grains were abnormal (Figure 7Q). Moreover, both the exine and intine were defective in *cslb03* mutants, with the exine and intine thickness of mature pollen being obviously reduced (Figure 7M,N). The pollen intine of the *cslb03* mutants was incomplete or even absent, which was also similar to that observed for the *bhlh010 bhlh089* mutants (Figure 1 and Figure 2). These results that *cslb03* mutants exhibited *bhlh010 bhlh089*–like pollen wall defective phenotypes, together with the results that *CSLB03* was directly activated by bHLH010, strongly demonstrated that the bHLH transcription factors act as a direct upstream regulator of *CSLB03* in the regulation of pollen wall morphogenesis.

## 3. Discussion

Pollen walls vary in surface morphology, whereas their fundamental structure is genetically conserved in both dicots and eudicots, being commonly composed of the exine and intine layers [3,4,8]. These two layers of the pollen wall are made up of distinct components. For instance, the exine consists of a robust material called sporopollenin that is derived from the tapetum, and the intine mostly consists of pectic polymers, cellulose, and hemicelluloses that are constructed from the microspores themselves [6,94,95,96].

Through thoroughly phenotypic, transcriptomic, and metabolomic analyses, we found that bHLH010 and bHLH089 not only are required for the development of anther tapetum but also directly regulate the development of pollen walls by regulating the expression of metabolic enzyme-encoding genes and transporter-encoding genes that are required for pollen intine and exine formation (Figure 8). Because the metabonomic detection requires a large number of biological samples and six biological replicates are required for each genotype, the small size of Arabidopsis anther could not meet such amount requirements. Therefore, we performed metabolomics analysis with an easily available inflorescence tissue in this study, although the inflorescence is a complex tissue with many tissue types and cell types. In order to uncover more anther-related data from the metabolome data, we jointly analyzed the metabolome data and the stages 4–7 anther transcriptome data. The following clues are demonstrated: First, the metabolites, such as polysaccharides, fatty acid derivatives, and flavonoids, in the *bhlh010*
*bhlh089* were significantly decreased (Figure 3), suggesting that bHLH010 and bHLH089 act as upstream regulators of these metabolic pathway. It is well known that the metabolites that make up the pollen wall are very complex. The components of the pollen walls may contain sporopollenin, sugar esters, fatty acid derivatives, pectin, cellulose, and other polysaccharides, which is consistent with our results (Figure 3E,F). Second, both combined transcriptome and metabolomics analysis and enrichment analysis showed that the activity of the same pathways was decreased. As shown in both the transcriptomic and qRT-PCR data, the expression of a number of genes involved in the metabolic pathways related to pollen wall development were found to be significantly downregulated in the *bhlh010 bhlh089* double mutant plants, including genes involved in the metabolic processes of polysaccharides, lipids, and metabolite transport (Figure 4 and Figure 5). Genes such as *CSLD4*, *ABCG26*, and *LAP6* are essential for pollen wall morphogenesis [32,81]. As upstream transcription factors, bHLH010 and bHLH089 likely activate the expression of these genes (Figure 6A–C). Other genes identified in this study may be involved in the function of pollen wall formation, which have not yet been reported. For example, *CSLD6, UGT76E2, and ABCG17* are likely to be important genes that regulate pollen wall morphogenesis. Third, transcriptional activation experiment results showed that the expressions of *CSLD5*, *CSLD6*, *LAP6*, and *UGT85A5* were activated by either bHLH010 or bHLH089. In addition, there were genes that were differentially activated by bHLH010 and bHLH089. *SUS3* and *CSLB03* were specifically induced by bHLH010, and *FRA8* and *TSM1* were specifically induced by bHLH089. Taking *CSLB03* as an example, further electrophoretic mobility shift assay (EMSA) experiments showed that bHLH010 directly bound to the promoter of *CSLB03* (Figure 6D–F). The expression pattern of *CSLB03* is similar to *bHLH010* in pollen. Semi-thin sections also showed that *CSLB03* was mainly expressed after the ninth stage of anther development (Appendix A). The *cslb03* mutant showed a similar phenotype of pollen wall defect as the *bhlh010 bhlh089* double mutant (Figure 7). In conclusion, bHLH010 and bHLH089 likely regulate the accumulation of metabolites and the formation of the pollen wall by regulating the expression of metabolic enzyme-encoding genes and transporter-encoding genes required for pollen intine and exine formation, including through the bHLH010-CLSB03 cascade (Figure 8). Overall, this study proposes a new regulatory model of how bHLHs regulate pollen wall development, which lays a foundation for plant male fertility and pollen wall morphogenesis research. 

In addition, as shown by the chemical dyes-staining analyses for the observation of the pollen intine and exine morphogenesis (Figure 2, Appendix A), the exine of WT pollen began to form at anther stage 8, followed by the intine layer at stage 10. However, both pollen wall layers of the *bhhlhh010 bhhlh089* plants was significant abnormalities: the *bhhlhh010 bhhlh089* pollen with an intact exine layer was hardly observed, and the pollen intine of a large portion of pollen was even not formed (Figure 2B). Those pollen grains lacking intine were malformed and resulted in pollen abortion. Interestingly, as long as the pollen intine is present, even if it is very thin and discontinuous, the pollen would develop into mature pollen grains with a shape similar to that of the WT ones, although there are substantial abnormalities in the pollen wall. Therefore, we speculate that the normal development of the pollen intine and exine is very important to keep the pollen morphology intact and mature, such as the severe abortion caused by the loss of the intine in *bhlh010 bhlh089* and *cslb03* mutant pollen (Figure 2 and Figure 7).

### 3.1. bHLH010 and bHLH089 May Regulate Pollen Wall Formation through the Synthesis of Polysaccharides

Pollen development and germination require much pectin and polysaccharide, and abnormal pectin biosynthesis also affects plant male fertility [97]. The pollen intine also comprises cellulose, pectin, and other polysaccharides. Cellulose synthase protein A (CESA) and a large number of cellulose synthase-like proteins (CSLs) constitute a large family. Both CESA and CSL have glycosyltransferase activity and are highly similar in both DNA sequences and the protein structures. However, CSL proteins do not have a zinc finger structure, which is the largest difference between CESA and CSL, meaning that CSL proteins have certain specificity. The *Arabidopsis* CSL genes contain nine distinct groups: the CSLA, B, C, D, E, F, G, H, and J subfamilies [98]. In *Arabidopsis thaliana*, *CSLD1* and *CSLD4* are specifically expressed in pollen and pollen tubes and are necessary for cellulose deposition and normal pollen tube growth [80,99]. *CSLD5* can also affect the formation of cell plates during the cell cycle and thus affects cell division [86,87]. However, the functions of *CSLD6* and *CSLE1* have not yet been reported (Figure 5A), and further studies are needed. The CSLB subfamily contains six family members, and CSLB04 is the closest CSLB family member to CSLB03, according to the phylogenetic analysis result (Appendix A), which is consistent with previous reports [98]. In this study, the genes of these cellulolytic synthase family members provide new research directions for the study of male fertility in plants.

In cellulose synthesis, in addition to the space composed of the cellulose synthase complex protein (CSC), which is formed by CESA and a variety of CSLs together, other enzymes are also required to synthesize cellulose. For example, UDPG produced by sucrose synthase 3 (SUS3) can be used as a direct substrate (Figure 5C). UDPG is the most abundant glycosylated donor in plants, and UDP-glycosyltransferase (UGT) can also directly use UDPG as a substrate for the synthesis of secondary metabolites, such as cellulose. UGT is also required for the polymerization of glycosylates to cellulose glycochains during cellulose synthesis [100], and UGT also plays a role in the synthesis of flavonols [101,102]. In the present study, *UGT85A5*, *UGT73B1*, *UGT73B3*, *UGT76C4*, *UGT76E2*, and *UGT91B1* were significantly downregulated (Figure 5B). However, there are few studies on these genes involved in pollen development, necessitating further study. These above results suggest that the transcription factors bHLH010 and bHLH089 directly regulate the expression of some genes related to pollen intine synthesis and subsequently control the formation of the pollen intine, which is consistent with the phenotype of *bhlh010 bhlh089* double mutant plants. TPS6 is highly expressed in the inflorescence tissue and is also expressed in both leaves and roots [103]. The mutant *csp-1*, which is defective in TPS6, has malformed leaf cells and reduced trichome branching [104]. However, whether TPS6 plays a role in anther development needs further study. The significant downregulation of TPS6 (Figure 5C) in the *bhlh010 bhlh089* mutant plants suggests that bHLH1010 and bHLH089 might affect pollen development by regulating *TPS6* expression in anthers.

### 3.2. bHLH010 and bHLH089 May Regulate Pollen Wall Formation through Transporters

The ABC transporter superfamily is one of the largest protein families discovered thus far, and by using the energy from ATP hydrolysis, its members can transport a variety of compounds, such as polysaccharides, lipids, and steroids [105]. There are 13 subfamilies in this family, among which ABCG family proteins compose the largest subfamily. ABCG transporter proteins contain two domains: a nucleotide-binding domain (NBD) and a transmembrane domain (TMD). At present, functions of several ABCG family proteins are reported, although further studies are needed to reveal their signaling pathways, especially their upstream regulators. The *Arabidopsis* ATP-binding transporters ABCG1 and ABCG16 affect reproductive development through the auxin signaling pathway [106]. ABCG9 and ABCG31 are mainly expressed in anther tapetal plasma membranes and microspores; these proteins can regulate the formation of the oil-bearing pollen coat of the pollen exine mainly through their involvement of the transport of sterol glycosides [107]. *ABCG26* is mainly expressed in anther tapetum, and was found to mediate the transport of polyketone and hydroxycinnamoyl spermidine to promote the formation of the pollen exine [37,39,42]. ABCG36 is located in the epidermal plasma membrane and interacts with calmodulin to enhance the disease resistance of plants [108]. In addition, the function of several *ABCG* members are demonstrated in rice. Os*ABCG15* was found to mainly express in the tapetal layer of anthers and is essential for anther development and pollen fertility in rice. The anthers and pollen grains of *Osabcg15* are severely aborted, and pollen walls are lacking [109]. These above results suggest that the ABCG family proteins likely play conserved roles in *Arabidopsis* and rice and transfer hydroxycinnamoyl spermidine to pollen grains to form the pollen exine. We found that the expression of *ABCG9*, *ABCG16*, *ABCG26*, *ABCG31*, and other genes were significantly downregulated in *bhlh010 bhlh089* double mutant plants (Figure 5D), and the pollen exine phenotype of mutants with defects in these genes was similar to that of the *bhlh010 bhlh089* double mutant. These results suggest that bHLHs may regulate the expression of these genes and thus regulate pollen morphogenesis. Further studies showed that bHLH089 could directly activate the expression of ABCG9 (Figure 6B), which might reflect a new pathway affecting the development of the pollen exine. Magnesium ion (Mg^2+^) transport plays a very important role in plant growth and development, reproductive development, and the improvement of plant resistance to stress. At present, some Mg^2+^ transport proteins in *Arabidopsis thaliana* have been found to play important roles in plant growth, development, and stress responses [110]. *MGT5* is specifically expressed in anthers, especially in anther tapetum and microsporocytes [111,112]. The expression of *MGT5* was significantly downregulated in *bhlh010 bhlh089* double mutant plants (Figure 5D). Therefore, the expression and function of *MGT5*, possibly also other MGT family members, is activated by the downstream of bHLH010/089 TFs, thereby regulating the cell-to-cell communication between tapetal cells and microsporocytes/microspores, as well as regulating the development of pollen.

## 4. Materials and Methods

### 4.1. Plant Material

The seeds of *Arabidopsis thaliana* WT, bhlh010 bhlh089 (T-DNA insertion), cslb03 (T-DNA insertion), mgt5 (T-DNA insertion), probHLH010:GUS, and probHLH089:GUS were sterilized and placed in culture dishes on Murashige and Skoog medium (MS). All *Arabidopsis thaliana* seeds had COL-0 background. The primers for detection of mutants are listed in Appendix A. The petri dish was placed in a light incubator for 10 days before the seedlings were transferred to the soil. The plants were grown in a daily cycle of 16 h of light at 22 °C and 8 h of darkness at 22 °C. The plant phenotypes were photographed with a Nikon D50 digital camera and a XiaoMi camera.

### 4.2. Generation of probHLH089: β-Glucuronidase (GUS) Transgenic Lines and GUS Staining Assays

We cloned the 2339-bp *bHLH089* promoter and inserted it into a pDONR intermediate vector through homologous recombination ultimately to generate a GUS-pGWB3 vector. The construct was transferred into *E. coli* DH5α and *Agrobacterium tumefaciens* GV3101 (P19), and the *Arabidopsis* plants were transformed using the floral-dip method [113]. *probHLH089:GUS* inflorescence tissues for GUS staining were harvested in the middle of the light period. Thirty-day-old transgenic inflorescences were submerged in GUS staining solution (0.1 M sodium phosphate buffer, 0.5 mM K_4_Fe(CN)_6_, 1 mM K_3_Fe(CN)_6_, 1% Triton X-100, 1 mM ethylenediaminetetraacetic acid (EDTA), and 1 mg/mL X-gluc) and incubated overnight in the dark at 37 °C. Phenotypes of GUS transgenic plants were photographed using an Axio Scope A1 (Zeiss, Oberkochen, Germany) with an AxioCam HRc camera (Zeiss) and a Discovery V8 dissecting stereomicroscope (Zeiss, Oberkochen, Germany, http://www.zeiss.com.cn/, accessed on 6 June 2022) equipped with a Spot Flex digital camera (Diagnostic Instruments, http://www.spotimaging.com/, accessed on 6 June 2022).

### 4.3. Semithin Section Examination and Pollen Wall Staining

Inflorescences of WT, *bhlh010 bhlh089* double mutant plants, and *probHLH089:GUS* transgenic plants were collected and fixed in an FAA solution (38% formaldehyde/acetic acid/50% ethanol =1:1:18). Different concentrations of ethanol were used for graded dehydration in an ethanol series (50%, 75%, 85%, 90%, and 100% [×2]), after which the sections were embedded in Technovit 7100 resin, as described previously. The Technovit resin-embedded sections were then sliced to a thickness of 0.2 µm using a motorized RM2265 rotary slicer (Leica, Wetzlar, Germany, http://www.leica-microsystems.com/, accessed on 6 June 2022) with a glass knife and then heat-fixed onto glass slides. The sections were stained with 0.05% toluidine blue for 30 min, rinsed, and then dried, after which they were imaged under an Axio Scope A1 microscope. For WT and *bhlh010 bhlh089* double mutant plant pollen wall staining examination, the sections were stained with 1% DODC and 10% calcofluor white in 0.1 M PBS buffer for 10 min. Images were taken under an Axio Scope A1 microscope (Zeiss) equipped with a 390–440 nm excitation filter and a 478 nm blocking filter

### 4.4. Scanning Electron Microscopy (SEM)

Anther and pollen were collected from the WT plants and *bhlh010 bhlh089* homozygous double mutant plants. Intact anthers of various stages were carefully cut from the inflorescence under an anatomical microscope and placed on conductive tape. The pollen was placed onto slides and subsequently covered with conductive tape. The pollen grains were sprayed with palladium and then observed with a Hitachi TM3000 scanning electron microscope.

### 4.5. Real-Time PCR

RNA extraction of inflorescence was performed with TRizol (TaKaRa RNAiso Plus, Kyoto, Japan), according to the manufacturer’s instructions. For each sample, 1 μg of total RNA was used for reverse transcription (RT) with the reverse transcription system (TaKaRa RR047A). The qPCR reaction was performed with TB Green^®^ Premix Ex Taq^TM^ II (Tii RnaseH Plus) (TaKaRa RR820A). The PCR program comprised an initial denaturation of 2 min at 95 °C and amplification by 35 cycles of 15 s at 95 °C and 1 min at 60 °C in a Bio-Rad CFX96 Touch. The actin gene was used as a control for normalization. The primer sequences used for real-time PCR analysis are provided in Appendix A.

### 4.6. Metabolome Extraction and Detection

The inflorescence tissues of WT and *bhlh010 bhlh089* were quantitatively weighed. About 50 mg of each sample was weighted out, and 400 μL methanol (containing 5 μg/mL 2-chloro-L-phenylalanine as internal standard) was added to it. The mixture was mixed by a vortex mixer for 1 min and homogenized for 3 min at 60 Hz two times. Then the mixture was centrifuged at 13,000× *g* rpm, 4 °C for 10 min. A supernatant was transferred to sampler vials for detection. An in-house quality control (QC) was prepared by mixing an equal amount of each sample. An Agilent 1290 Infinity II UHPLC system coupled to an Agilent 6545 UHD and Accurate-Mass Q-TOF/MS was used for LC–MS analysis. The chromatographic column used was Waters Waters XSelect^®^ HSS T3 (2.5 μm 100 × 2.1 mm). Mobile phase: A: aqueous solution with 0.1% formic acid. B: acetonitrile solution with 0.1% formic acid. Flow rate: 0.4 mL/min. Column temperature: 40 °C. Injection volume: 4 μL. Gradient elution condition optimized: 0–3 min, 20% B; 3–9 min, 20–95% B; 9–13 min, 95% B; 13–13.1 min, 95–5% B; 13.1–16 min, 5% B. Mass spectrometry was operated in both positive and negative ion modes. The parameters optimized were as follows: Capillary voltage: 3.5 kV. Drying gas flow: 10 L/min. Gas temperature: 325 °C. Nebulizer pressure, 20 psig. Fragmentor voltage: 120 V. Skimmer voltage: 45 V. Mass range: *m*/*z* 100–1500.

### 4.7. Differential Metabolite Analysis

Raw data were converted into the common (mz.data) format by the Agilent Masshunter Qualitative Analysis B.08.00 software (Agilent Technologies, Palo Alto, CA, USA). In the R software platform, the XCMS program 3.18.0 (CA, USA) was used in peak identification, retention time correction, and automatic integration pretreatment. Then the data were subjected to internal standard normalization. Visualization matrices containing a sample name, *m*/*z*-RT pair, and peak area were obtained. A total of 4345 features were in positive mode and 3110 features in negative mode. After editing, the data matrices were imported into SIMCA-P 14.1 (Umetrics, Umea, Sweden), mean-centered, and scaled to Pareto variance. Then, multivariate analysis was conducted. Differential metabolites were screened out by the VIP (variable importance in projection) value of OPLS-DA model (VIP ≥ 1) and independent sample t-test (*p* < 0.05). A qualitative method of differential metabolites is searching in the online database for accurate molecular weight comparison. Adduct manner: [M+H]^+^ and [M+Na]^+^ was selected in positive mode, [M−H]^−^ in negative mode. Mass error value: 30 PPM. Differential metabolites identified are shown in Appendix A.

### 4.8. Analysis of the bhlh010 bhlh089 Transcriptomic Data

The anther transcriptomic raw data of the WT and *bhlh010 bhlh089* double mutants [79] were reanalyzed with the following procedures: First, the sequencing reads were quality trimmed using TrimGalore 0.6.6 (Cambridge, UK)(https://github.com/FelixKrueger/TrimGalore, accessed on 8 March 2022), then aligned to the TAIR10 genome using STAR 2.7.7a (New York, NY, USA) (https://github.com/alexdobin/STAR, accessed on 8 March 2022) with default settings. Then, the reads that mapped to unique positions were counted using the feature Counts 2.0.1 (Victoria, Australia) [114]. After that, samples in each treatment group were clustered using principal component analysis (PCA), and the differentially expressed genes (DEGs) between the different treatment groups were identified by the DESeq R package with a *p*-value < 0.01 and |log_2_ (fold change, FC)| > 1. Finally, the heatmap was performed to determine the expression pattern of DEGs under different experimental conditions. Differential genes identified are shown in Appendix A.

### 4.9. Combined Metabolome and Transcriptome Analysis

GO enrichment and KEGG pathway analysis were implemented to screen the DEGs that were significantly enriched in GO terms and metabolic pathways at *p* < 0.01 using the topGO R packages and KOBAS 3.0 software (Beijing, China) (http://www.genome.jp/kegg/, accessed on 6 March 2022), respectively. The differential accumulated metabolites’ (DAMs) R package with a *p*-value < 0.05 and |log_2_ (fold change, FC)| > 1. Differential genes and metabolites were combined, analyzed using the MetaboAnalyst website (Quebec, Canada) (https://www.metaboanalyst.ca/MetaboAnalyst/upload/JointUploadView.xhtml, accessed on 8 March 2022).

### 4.10. Dual-Luciferase Reporter Assays

We amplified the promoter of *CSLB03*, *CSLD5*, and other genes from WT plants with Vazyme 2× Phanta^®^ Max Master Mix DNA polymerase (P515-01, Nanjing, China). Then, the promoter inserted it into the pGreenII 0800-LUC vector. We also amplified and cloned the genes of bHLH010 and bHLH089 into the modified 3FLAG-C3F vector (Gateway, Thermo Fisher, Waltham, MA, USA). Primer sequences are provided in Appendix A. The construct was transferred into DH5α (WeiDi, DL1001M, Shanghai, China) and *Agrobacterium tumefaciens* GV3101 (P19) (WeiDi, AC1001M, Shanghai, China). After confirmation by sequencing, the constructs were transformed into *Agrobacterium* GV3101. Then, *Agrobacterium* GV3101 was mixed and injected into the *Nicotiana. benthamiana* leaves. After 48 h growth under a 16 h light/8 h dark, we used a YEASEN dual-luciferase reporter gene assay kit (11402ES60, Shanghai, China) and a Synergy™ 2 microplate spectrophotometer (Biotek, Winooski, VT, USA) to detect the luminescence values and statistics.

### 4.11. Electrophoretic Mobility Shift Assay

The EMSA was performed according to previously reported methods. The full-length coding sequence of the *bHLH010* gene was amplified and cloned into the pSUMO-HIS vector to produce the HIS-binding protein–bHLH010 construct. Primer sequences are provided in Appendix A. The constructed vector was transferred into Rosetta (DE3) *E. coli* for prokaryotic expression. Monoclonal positive bacteria were selected and added to 100 mL LB (Luria-Bertani) (0.5 g yeast extract, 1 g peptone, 1 g NaCl, 100 mL distilled water, pH 7.0) liquid medium (37 °C, 3 h). Then IPTG (isopropyl β-d-thiogalactoside) was used to induce the expression of fusion protein at the concentration of bacteria liquid (OD600 = 0.6). bHLH010-HIS fusion protein was induced overnight at 18 °C (10–12 h, IPTG 0.2 mm), and the extracted protein could be stored at −80 °C. The DNA fragment including the E-box in the *CSLB03* region was generated using specific primers to produce 5′ biotin-labelled and unlabeled competitor probes. Primer sequences are provided in Appendix A. The EMSA assay was performed with a LightShift Chemiluminescent EMSA Kit (20148, Thermo Fisher). The images were captured using a CLiNX Chemiluminescent Imaging System (https://www.clinx.cn/, accessed on 6 June 2022).

### 4.12. In Vitro Pollen Germination Analysis

Pollen grains of the wild-type (col-0) and *bhlh010 bhlh089* plants were germinated on the standard agar medium containing 5 mM Ca^2+^ (with an equal molar ratio of CaCl_2_ and Ca[NO_3_]_2_) and incubated at 22 °C for 6 h before observation. In this case, all pollen that produced visible pollen tubes was considered successfully germinated pollen. Subsequent statistical analysis on germination rate was also based on this standard. 

### 4.13. Phylogenetic Analysis

To investigate whether the *CSLB03* gene has homologous genes, we constructed an evolutionary tree of the CSLB subfamily. Protein sequences of homologs of CLSB03 were obtained by blast+ (http://blast.ncbi.nlm.nih.gov/Blast.cgi, accessed on 2 September 2022) from a database that contained six rosid plant protein sequences, and aligned using MAFFT. A maximum likelihood tree was constructed using FastTree [115,116].

## Figures and Tables

**Figure 1 ijms-23-11683-f001:**
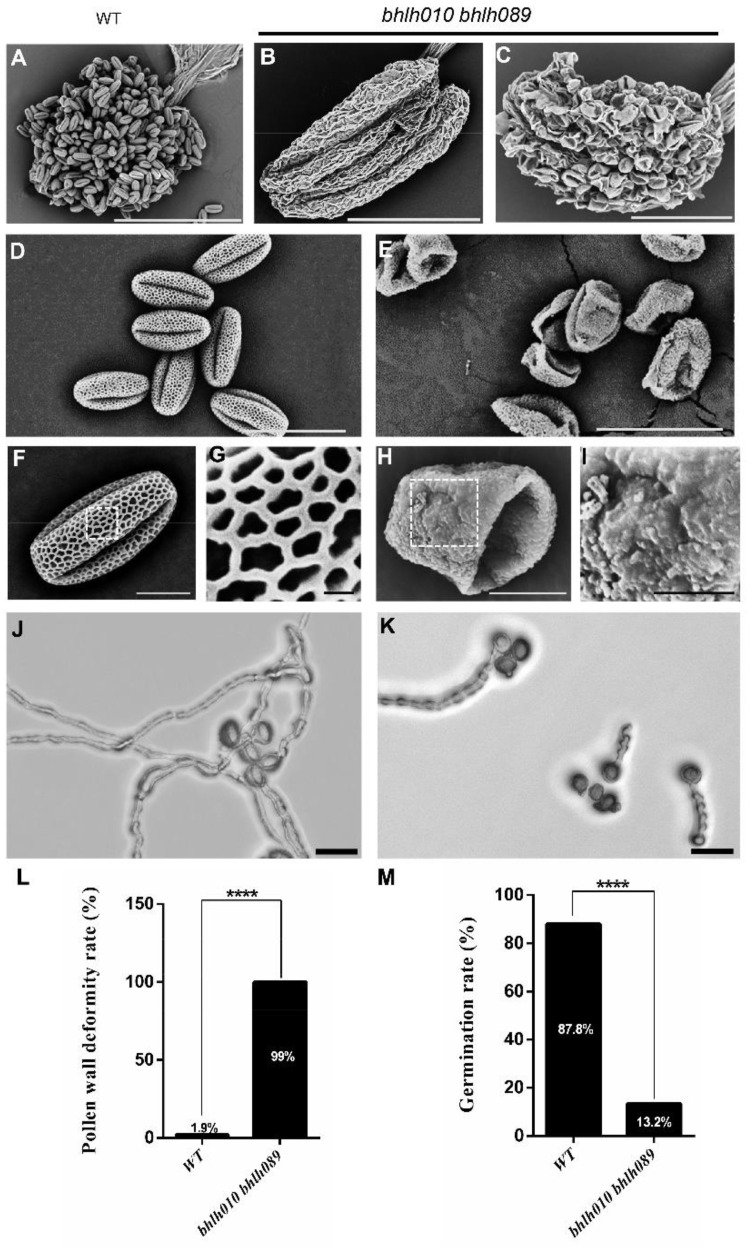
The pollen development of a *bhlh010 bhlh089* double mutant was severely malformed. (**A**–**C**) The anthers of WT and *bhlh010 bhlh089* double mutants were observed by scanning electron microscopy (SEM). (**A**) The anther of WT flower. Bar = 200 μm. (**B**) The anther of the *bhlh010 bhlh089* early flower. Bar = 200 μm. (**C**) The anther of *bhlh010 bhlh089* late flowers. Bar = 100 μm. (**D**–**I**) SEM of pollen gains of the WT (**D**) and *bhlh010 bhlh089* (**E**) pollen. (**F**,**H**) Pollen exine structure of WT (**F**) and *bhlh010 bhlh089* (**H**). Bars = 20 μm. (**G**,**I**) SEM images of enlarged exine surface of WT (**G**) and *bhlh010 bhlh089* (**I**) pollen. Bars = 10 μm. (**J**,**K**) Pollen germination of the WT (**J**) and *bhlh010 bhlh089* (**K**) plants. Bars = 50 μm. (**L**,**M**) Statistic analysis of the rate of pollen with defective exine phenotype (**L**) and the germination rate (**M**) of the WT and *bhlh010 bhlh089* plants. The asterisks in (**L**,**M**) indicate a statistically significant difference from the WT control (****, *p* < 0.0001, Student’s *t*-test).

**Figure 2 ijms-23-11683-f002:**
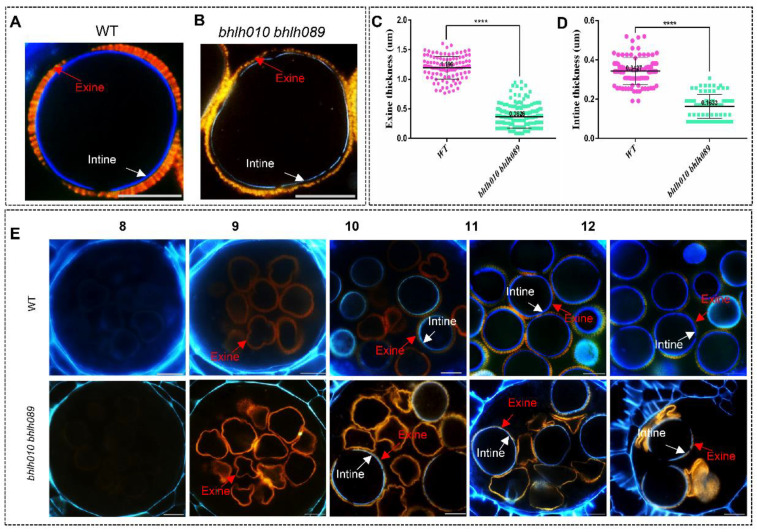
Pollen wall formation and differences of WT and *bhlh010 bhlh089* double mutant plants. (**A**,**B**) Schematic diagram of differences in the pollen wall structure. (**A**) The exine and intine of pollen of WT are relatively uniform. (**B**) The pollen exine of the *bhlh010 bhlh089* double mutant plants was uneven and partially missing, and the pollen intine was thin and discontinuous. Some pollen does not have an intine. (**C**,**D**) The thicknesses of the pollen exine and the pollen intine were calculated. The exine (**C**) and intine (**D**) of the pollen of WT are relatively uniform, with thicknesses of 1.196 ± 0.01813 μm (*n* = 110) and 0.3626 ± 0.01420 μm (*n* = 179), respectively. The pollen exine (**C**) and intine (**D**) of the double mutant plants are not uniform, and the pollen intine is thin, with thicknesses of 0.3427 ± 0.006770 μm (*n* = 99) and 0.1633 ± 0.006480 μm, (*n* = 87), respectively. Only the intine thickness of pollen with the intine is counted. The asterisks in (**C**,**D**) indicate a statistically significant difference from the WT control (****, *p* < 0.0001, Student’s *t*-test). (**E**) Transverse sections of the pollen wall development were performed in WT and the *bhlh010 bhlh089* double mutant. There was an obvious difference between WT and the *bhlh010 bhlh089* double mutant in stages 8–12. The white arrow indicates the intine, and the red arrow indicates the exine. Bars = 10 μm. The sections of the pollen wall development at stage 5–7 of WT and the *bhlh010 bhlh089* double mutant in Appendix A.

**Figure 3 ijms-23-11683-f003:**
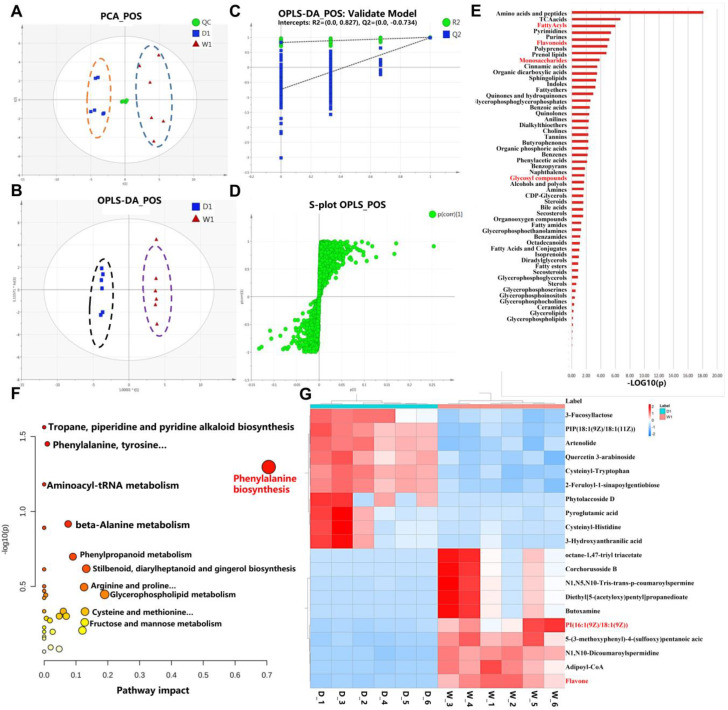
Untargeted metabolomics analysis of differential metabolites in inflorescences of WT and *bhlh010 bhlh089* double mutant. (**A**–**D**) PCA, OPLS-DA analysis, and system stability investigation. PCA is an unsupervised model analysis method, and it can reflect the real difference among groups more reliably and eliminate outliers. (**A**) The overall PCA scores plots. The yellow dotted line is the *bhlh010 bhlh089* (D1, blue square) double mutant inflorescence sample, and the blue dotted line is the WT (W1, red triangle) inflorescence sample. (**B**) OPLS-DA score plots. The black dotted line is the *bhlh010 bhlh089* (D1) double mutant inflorescence sample, and the purple dotted line is the WT (W1) inflorescence sample. In the model of positive mode, R2X = 0.716, R2Y = 0.999, Q2 = 0.987. The model is thought to be reliable if these parameter values > 0.4. (**C**) OPLS-DA models were validated by permutation tests to access whether the modes were overfit. R2 = 0.827, Q2 = −0.734. (**D**) S-plots of Group W1 and Group D1. The points far from the origin contribute to intergroup difference significantly and have lager VIP values. The above (**A**–**D**) horizontal and vertical coordinates have no practical significance. The above only shows the results of positive ion accumulation mode. (**E**) Cluster analysis of total differential metabolites. Differential metabolites were screened out by the VIP (variable importance in projection) value of the OPLS-DA model (VIP ≥ 1) and independent sample *t*-test (*p* < 0.05). Adduct manner: [M+H] ^+^ and [M+Na]^+^ were selected in positive mode, [M-H]^−^ in negative mode. Mass error value: 30 PPM. (**F**) The bubble plot of significantly reduced metabolic pathways by the metabolomics of *bhlh010 bhlh089* double mutant inflorescence. Each circle denotes a metabolic pathway, with the red color indicating a higher −log (*p*) value and the yellow color indicating a lower −log (*p*) value. The size of the circle indicates the pathway impact value in the topological analysis. (**G**) The top 10 differential metabolites were increase and decrease in WT (W1) and *bhlh010 bhlh089* double mutant (D1) inflorescence ranking. The red words in E, F and G represent metabolic pathways and metabolites that might be involved in pollen development.

**Figure 4 ijms-23-11683-f004:**
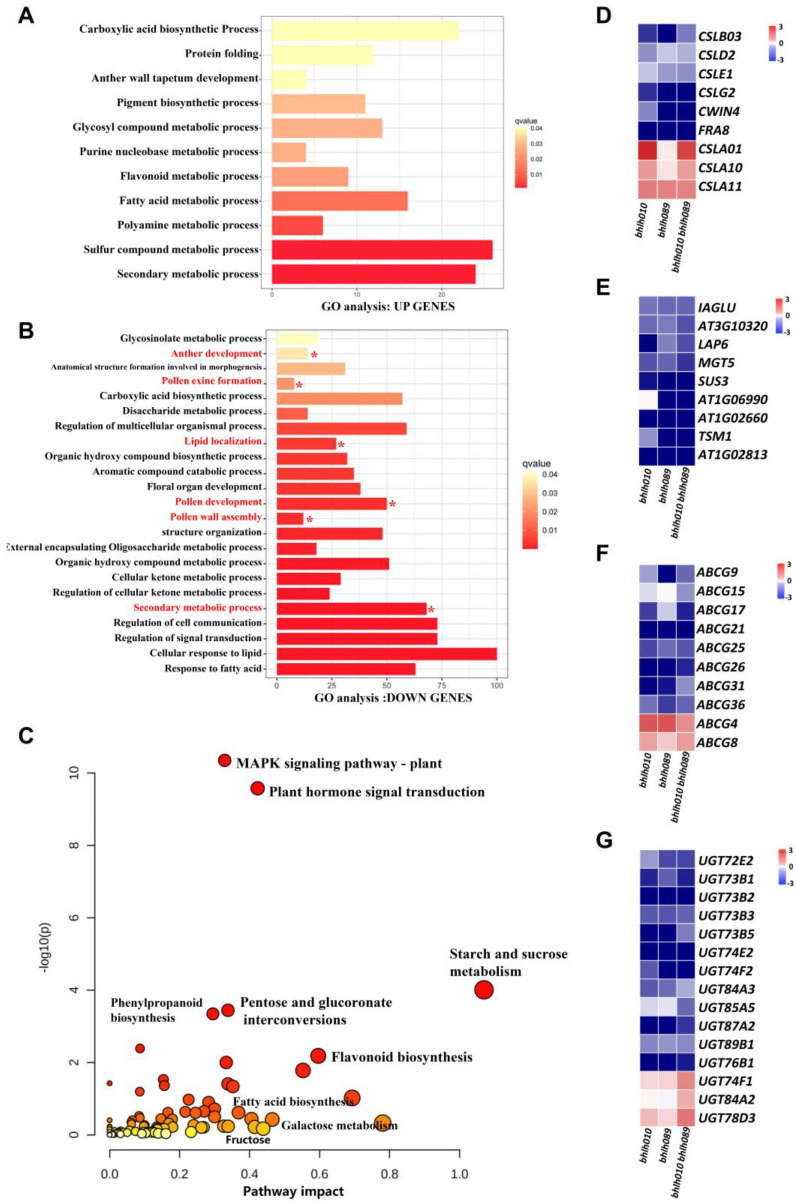
Combined untargeted metabolomic and transcriptomic analysis. (**A**,**B**) Cluster analysis of transcriptome data of WT and anther of double mutant plants. (**A**) Upregulation of gene-related pathways in the anthers of the *bhlh010 bhlh089* double mutant plants compared with the wild type. (**B**) Downregulation of gene-related pathways in the anthers of the *bhlh010 bhlh089* double mutant plants compared with the wild type. Asterisks indicate pathways associated with pollen development. The horizontal axis shows the number of genes, and the darker the ruler, the greater the correlation. (**C**) Combined metabolomics and transcriptome analysis of WT and *bhlh010 bhlh089* double mutants. Each circle denotes a metabolic pathway. The bigger the circle, the more metabolites and genes are involved. The redder the color, the greater the correlation. (**D**–**G**) Significantly different genes in WT and *bhlh010 bhlh089* double mutant anthers. (**D**) Differential genes associated with cellulose synthesis. (**E**) Other metabolism-related differential genes. (**F**) Differential genes associated with metabolite transport. (**G**) Differential genes associated with glycosyltransferase.

**Figure 5 ijms-23-11683-f005:**
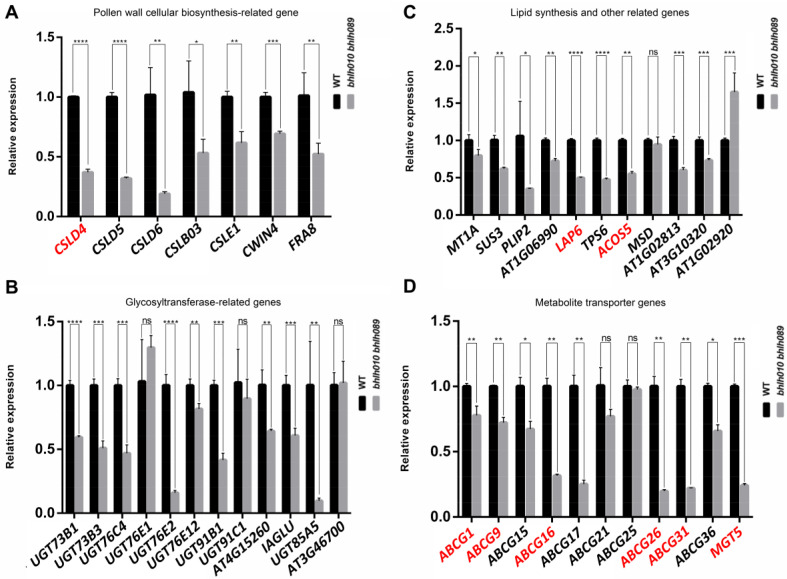
Detection of the expression of genes related to bHLHs that may regulate pollen wall development. (**A**–**D**) The expressions of pollen wall development-related genes were examined in the WT and the *bhlh010 bhlh089* double mutants. (**A**) Different genes related to pollen wall cellulose synthesis were detected by qRT-PCR. (**B**) Different genes related to glycosyltransferase-related genes were detected by qRT-PCR. (**C**) Different genes related to lipid synthesis and other pathways were detected by qRT-PCR. (**D**) Different genes related to metabolite transport were detected by qRT-PCR. The red font indicates validated genes associated with pollen wall development. Actin gene served as the reference. The asterisks indicate a statistically significant difference (*, *p* < 0.05; **, *p* < 0.01; ***, *p* < 0.001; ****, *p* < 0.0001, ns indicates no significant difference). Error bars indicate SD. Experiments were performed at least three independent times. Genes marked in red have been reported to be involved in pollen wall formation.

**Figure 6 ijms-23-11683-f006:**
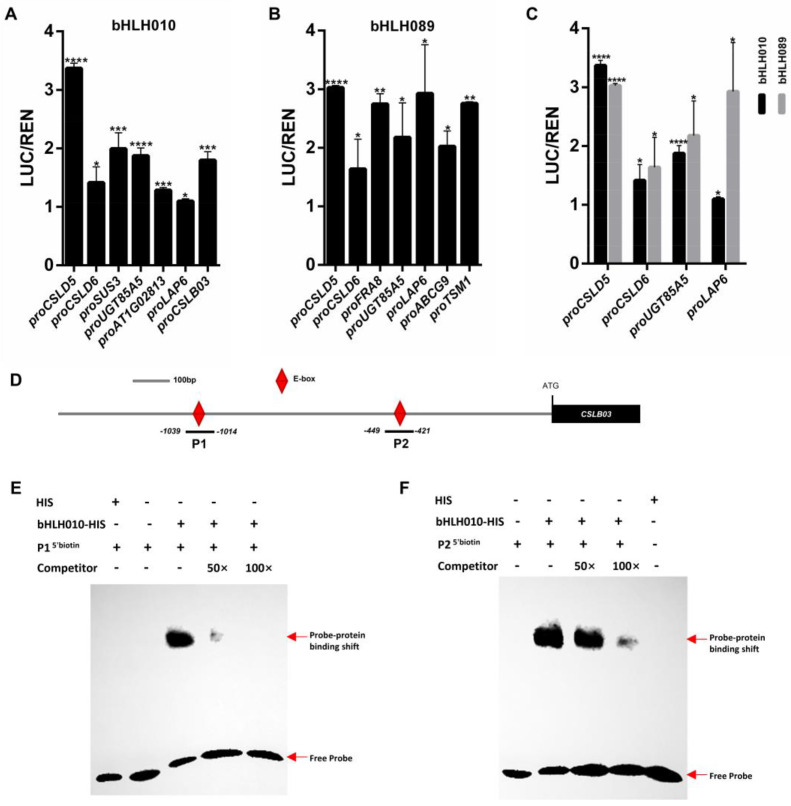
bHLH010 activates the expression of *CSLB03* and directly binds to the two E-boxes at the *CSLB03* promoter. (**A**–**C**) Dual-luciferase assay was used to detect the transcriptional activation activity of bHLH010 and bHLH089 to downstream candidates. (**A**) The transcriptional activation activity of bHLH010 to downstream candidates. (**B**) The transcriptional activation activity of bHLH089 to downstream candidates. (**C**) The four genes that could be activated by both bHLH010 and bHLH089 are shown. Significant differences compared with WT were determined using Student’s *t*-test, which was used for difference comparison. *, *p* < 0.05; **, *p* < 0.01; ***, *p* < 0.001; ****, *p* < 0.0001. Data were means ± SD (*n* = 3). (**D**) The *CSLB03* promoter region (grey) has two predicted E-boxes (red rhombus). (**E**,**F**) EMSA of the binding of bHLH010 TF to the P1 (**E**) and P2 (**F**) E-box in the *CSLB03* promoter region. The binding bands and the free probes are indicated with red arrows.

**Figure 7 ijms-23-11683-f007:**
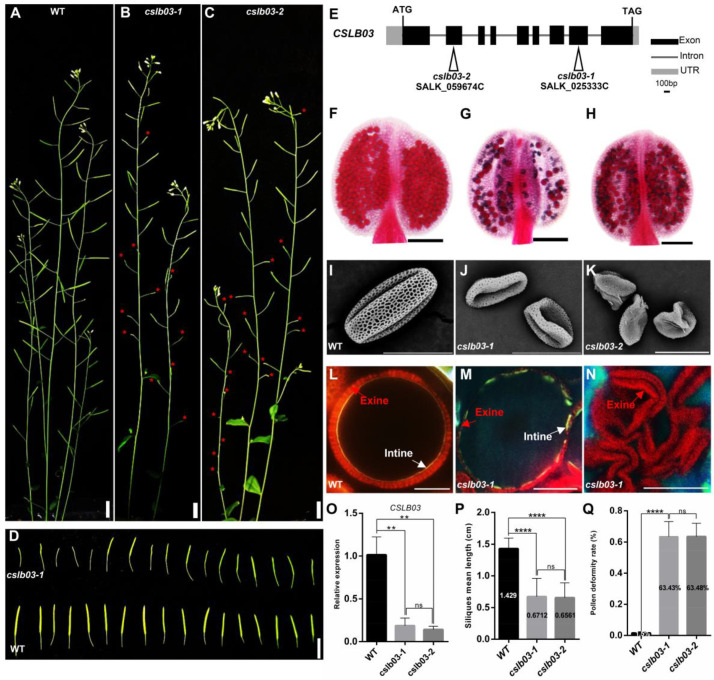
Pollen developmental phenotypes of *cslb03* mutant plants. (**A**–**C**) Plant fertility of the WT (**A**), *cslb03-1* (**B**), and *cslb03-2* (**C**) plants. Red asterisks indicate severely aborted siliques. Bar = 1 cm. (**D**) Silique phenotypes of the WT and *cslb03-1* mutant plants. Bar = 1 cm. (**E**) Schematic diagram of the T-DNA insertion sites of the *cslb03-1* and *cslb03-2* mutants. (**F**–**H**) Alexander staining of anthers of the WT (**F**), *cslb03-1* (**G**), and *cslb03-2* (**H**) plants. Bars =100 μm. (**I**–**K**) SEM image of the WT (**I**), *cslb03-1* (**J**), and *cslb03-2* (**K**) pollen grains. Bar = 20 μm. (**L**–**N**) Staining of the exine and intine structure of pollen grains from the WT (**L**) and *cslb03-1* (**M**,**N**) plants. Bars =10 μm. (**O**) The relative expression levels of the *CSLB03* gene in the WT, *cslb03-1*, and *cslb03-2* inflorescence. (**P**) The average length of siliques from the WT, *cslb03-1*, and *cslb03-2* mutant plants. (**Q**) The rate of defective pollen in the WT, *cslb03-1*, and *cslb03-2* anthers. Values of (**O**) are mean ± SD of three independent experiments. Values of (**P**,**Q**) are mean ± SD of more than three independent plants. The asterisks indicate a statistically significant difference (**, *p* < 0.01; ****, *p* < 0.0001, ns indicates no significant difference, Student’s *t*-test).

**Figure 8 ijms-23-11683-f008:**
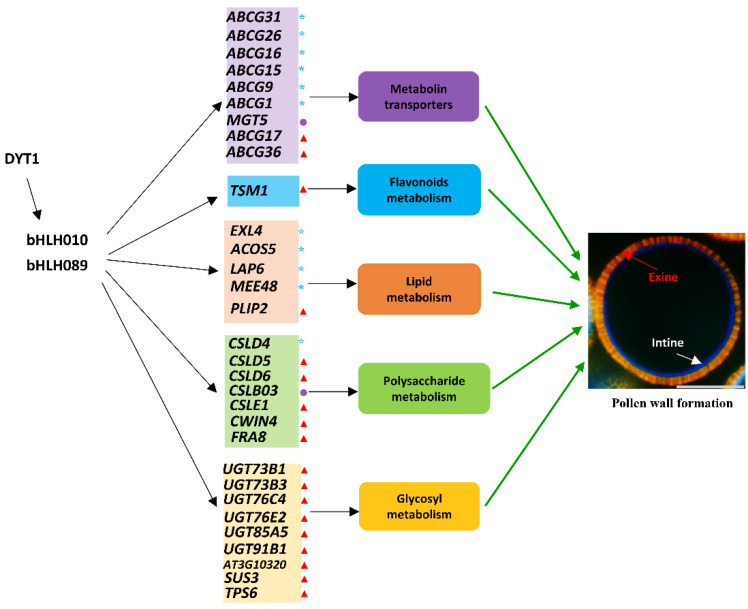
A proposed model of how bHLH transcription factors regulate pollen wall development. Based on transcriptome and metabolome data of *bhlh010 bhlh089*, the transcript level of enzymes and transporters was significantly downregulated. Metabolism-related enzymes and transporters affect pollen wall morphogenesis by regulating the metabolic accumulation of polysaccharides, lipids, flavonols, and other substances. In this study, we identified these new candidate genes that may regulate pollen wall morphogenesis. The blue asterisks indicate genes associated with the development of the exine, the purple circles indicate genes associated with the development of the intine, and the red triangles indicate genes that may be involved in the development of a pollen wall.

## Data Availability

*Arabidopsis* sequence data in this article can be found in the *Arabidopsis* Information Resource (TAIR, http://www.Arabidopsis.org/, accessed on 6 September 2022) under accession nos. *bHLH010* (AT2G31220), *bHLH089* (AT1G06170), *CSLB03* (AT2G32530), *ABCG9* (AT4G27420), *CSLD4* (AT4G38190), *CSLD5* (AT1G02730), *CSLD6* (AT1G32180), *SUS3* (AT4G02280), *FRA8* (AT2G28110), *ABCG15* (AT3G21090), *ABCG17* (AT3G55100), *ABCG21* (AT3G25620), *ABCG26* (AT3G13220), *MGT5* (AT4G28580), LAP6 (AT1G02050), *CWIN4* (AT2G36190), *TPS6* (AT1G68020), C*SLE1* (AT1G55850), *IAGLU* (AT4G15550), *UGT76E2* (AT5G59590), *UGT73B1* (AT4G34138), *UGT73B2* (AT4G34135), *UGT76C4* (AT5G05880), *UGT91C1* (AT5G49690), *UGT91B1* (AT5G65550). The original metabolic data from this article has been submitted to MetaboLights database under accession number MTBLS5719 (https://www.ebi.ac.uk/metabolights/presubmit, accessed on 6 September 2022).

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
