# Peer review of "bHLH010/089 Transcription Factors Control Pollen Wall Development via Specific Transcriptional and Metabolic Networks in Arabidopsis thaliana"

_ijms, 2022, doi:10.3390/ijms231911683_

Round 1

Reviewer 1 Report

The plant pollen wall protects the male gametophyte from various biotic and abiotic stresses. The formation of a unique pollen wall structure and elaborate exine and intine pattern is a well-organized process. The Arabidopsis bHLH010/089 (basic helix-loop-helix) genes are functionally redundant and are required for both anther development and normal expression of DYT1-activated anther-related genes. However, their function in pollen wall development remains largely unknown. In the current study, at the cytological, metabolomic and transcriptomic levels, the authors demonstrated that the bHLH010 and bHLH089 TFs together play significant roles in regulating the development of both pollen exine and intine, through regulating the expression of metabolic enzyme-encoding and transporter-encoding genes, as well as affecting the accumulation of polysaccharides, cellulose, flavonols and other metabolites that required for the formation of pollen walls. Moreover, they found CSLB03 is a novel target of bHLH10 with no biological function. Their results provided a certain basis for subsequent studies on male fertility. The overall level of this manuscript is good. The cytological and molecular biology experiments were pretty clear and convictive , however, the figures of the metabolomic and transcriptomic analysis in the manuscript are not very clear.

1.       It is recommended to increase the font size of the ordinates and x-coordinate label in Figure 1L and Figure 2C and D, Figure 3A-F. Increase the font size of Figure 3 A-D.

2.       Line240: “upregulated (-log[fold change (FC)]≥1 and P<0.01)”,why “-“?

3.       Line 627 and 633: The stander of DEGs screening is “P-value < 0.01 and log2 (foldchange, FC) > 1”, the description should be “P-value < 0.01 and |log2 (foldchange, FC)| > 1”.

4.       In Figure 4D-G:  the data of bhlh010, bhlh089, bhlh010 bhlh089 derived from both metabolomic and transcriptomic analysis? TableS2 only showed the expression data of WT and double mutants for metabolomic analysis, while TableS3 only showed data of WT and bhlh089. Do they from reference 79?

5.       CSLB03 encodes a gene similar to cellulose synthase, cellulose synthase is important for plant cell wall (including the pollen wall). According to reference 79 and the present study, the extent of decrease in length of siliques compare to WT seems a little strong in cslb03 than that in bhlh010 bhlh089 double mutants. Do the author has any idea about this case?

6.       Line359: “cslb03-1 (M-M) plants” should be “M-N”.

7.       Arabidopsis thaliana” should use italics.

8.       Please use a unified expression of “quantitative real-time PCR (qRT-PCR)” in the whole manuscript.

Author Response

  1. It is recommended to increase the font size of the ordinates and x-coordinate label in Figure 1L and Figure 2C and D, Figure 3A-F. Increase the font size of Figure 3 A-D.

Response: Thanks. We have increased the font size of the ordinates and x-coordinate label in Figure 1L and Figure 2C and D, Figure 3A-F; and increase the font size of Figure 3 A-D as well.

  1. Line240: “upregulated (-log[fold change (FC)]≥1 and P<0.01)”,why “-“?

Response: The “-“has been deleted in the revised version.

  1. Line 627 and 633: The stander of DEGs screening is “P-value < 0.01 and log2 (foldchange, FC) > 1”, the description should be “P-value < 0.01 and |log2 (foldchange, FC)| > 1”.

Response: Thanks. The description has been changed to “P-value < 0.01 and |log2 (foldchange, FC)| > 1” in Line574-575 and Line582.

  1. In Figure 4D-G:  the data of bhlh010, bhlh089, bhlh010 bhlh089 derived from both metabolomic and transcriptomic analysis? Table S2 only showed the expression data of WT and double mutants for metabolomic analysis, while Table S3 only showed data of WT and bhlh089. Do they from reference 79?

Response: Thank you for this curious. And we also apologize for not describing them very clearly in the previous version.

In Figure 4D-G:  the data of bhlh010, bhlh089, bhlh010 bhlh089 shown in this figure derived from only transcriptomic analysis.

Table S2 showed the different accumulation of metabolites of the WT and the bhlh010 bhlh089 double mutant, which was achieved novelly in this work.

Table S3 showed the gene expression differences in the anthers of WT and bhlh010 bhlh089 double mutant, which were obtained through reanalyzing of the original transcriptomic raw data that were previously generated in our lab and published with reference 79.

The discription about the bhlh010 bhlh089 transcriptomic data have been added in line 214-219 of Page 6 (the Result section), and line 567-569 of page 15 (the Materials and Methods section).

  1. CSLB03 encodes a gene similar to cellulose synthase, cellulose synthase is important for plant cell wall (including the pollen wall). According to reference 79 and the present study, the extent of decrease in length of siliques (compare to WT)seems a little strong in cslb03 than that in bhlh010 bhlh089 double mutants. Do the author has any idea about this case?

Response: CSLB03 (CELLULOSE SYNTHASE-LIKE B3) encodes a gene similar to cellulose synthase, but its function is not clear yet.

According to our previous (reference 79) and current study (this paper), both bhlh010 bhlh089 and cslb03 produced siliques with decreased length compared to that of the WT. However, since the phenotypic data in the previous paper (reference 79) and our current work are not from the same batch of plants, the quantitative comparison between them is not rigorous.

Therefore, we planted the WT, bhlh010 bhlh089, and cslb03 mutants in the same batch and performed comparative phenotypic analysis.  The results showed that the average silique length of the WT, bhlh010 bhlh089,  cslb03-1 and cslb03-2 plants were 1.43 cm, 0.45 cm (Figure S1),  0.67 cm and 0.66 cm (Figure 7P), respectively. These results indicating that the siliques length of both bhlh010 bhlh089 and cslb03 were decreased, and  bhlh010 bhlh089 decreased more severely.

  1. Line359: “cslb03-1 (M-M) plants” should be “M-N”.

Response: Thanks. The description has been changed to be “M-N” in the legend of figure 7, line 344 of the revised version (Line 327 of the last version).

  1. “Arabidopsis thaliana” should use italics.

Response: Italics have been used for all “Arabidopsis thaliana” in the whole manuscript.

  1. Please use a unified expression of “quantitative real-time PCR (qRT-PCR)” in the whole manuscript.

Response: Thanks. All relevant expression have been unified into “quantitative real-time PCR (qRT-PCR)”.

Reviewer 2 Report

The authors in this study examine the role of two bHLH transcription factors bHLH010 and bHLH089 in pollen wall development. They use a plethora of approaches to answer their scientific questions like reverse genetics, morphology and microscopy, metabolomics and transcriptomics.  

Some comments about the manuscript

-the introduction is well written and provides a good basic description of the field

-the figures as also well presented though the font size should increase in some places to be easier to read

-in some figure legends the authors describe their results, this is not necessary, the legends should clearly describe the method and the material used to generate the data, the description should be in the main text

-the manuscript could benefit if the authors provide a short explaination of what the data suggest at the end of the description of each experiment

-Fig1 K and M, the authors should state how they define the successful germination, usually the germinated pollen is classified in stages,  that way a better picture of the germination process is provided

-the authors should describe the phenotype of the single mutants, so that the phenotype of the double could be better understood

-line 44 hydrolytic enzymes

-line 163 counted not counte

-the authors used inflorescence tissue for both transcriptomic and metabolomic analyses, though easily available inflorescence is a complex tissue with many tissue types and cell types of which only the pollen was of interest, the authors should address this issue in the discussion

-line 582 why 50 cycles in the real time PCR? isn't that too high?

-in material and methods the authors should provide more information on the clonings and the protein expression methods used

-though the authors state that there are no information about the CSLB03 gene, the sequence analysis did not provide any clues about its function? its homology with other genes?

Author Response

Comments and Suggestions for Authors

The authors in this study examine the role of two bHLH transcription factors bHLH010 and bHLH089 in pollen wall development. They use a plethora of approaches to answer their scientific questions like reverse genetics, morphology and microscopy, metabolomics and transcriptomics.  

Some comments about the manuscript

-the introduction is well written and provides a good basic description of the field

Response: Thank you for the positive comment.

-the figures as also well presented though the font size should increase in some places to be easier to read

Response: Thanks. We have increased the font size of the ordinates and x-coordinate label in Figure 1L and Figure 2C and D, Figure 3A-F; and increase the font size of Figure 3 A-D as well.

-in some figure legends the authors describe their results, this is not necessary, the legends should clearly describe the method and the material used to generate the data, the description should be in the main text

Response: The figure legends have been double checked and revised following your suggestion.

The description “The pollen of the bhlh010 bhlh089 double mutants were small and irregular. The exine of pollen has defects, and some pollen have no intine at all.” has been deleted from Figure 2;

and the following sentences “All QC samples were gathered well and the dispersion of QC samples was obviously lower than samples to be detected, indicating the system stability was fine.”, “There is an obvious separating trend between groups W1 and D1.”, “The models were not overfit. None of the above horizontal and vertical axes have any meaning, this is a way of showing the descending dimension of three-dimensional space”, and “The above results indicated that there were significant differences between W1 group and D1 group.” have been removed from the legend of Figure 3.

-the manuscript could benefit if the authors provide a short explaination of what the data suggest at the end of the description of each experiment.

Response: The manuscript has been revised as suggested.

-Fig1 K and M, the authors should state how they define the successful germination, usually the germinated pollen is classified in stages,  that way a better picture of the germination process is provided.

Response: Thanks. Pollen grains of the wild-type (col-0) and bhlh010 bhlh089 plants were incubated on the germination medium and incubated at 22℃ for 6 hours before observation. In this case, all pollen that produced visible pollen tubes is considered as successfully germinated pollen. Subsequent statistic analysis on germination rate was also based on this standard. These above description has been added to 4.11 of the method section, Line 611-616 of page 16.

-the authors should describe the phenotype of the single mutants, so that the phenotype of the double could be better understood

Response: Thank you. Neither bhlh010 nor bhlh089 single mutant showed obvious fertility defects; whereas the bhlh010 bhlh089 double mutants showed defective anther development, with obviously reduced anther size and pollen amount [75]. In addition, we also found that the silique length of bhlh010 bhlh089 plants was also obviously decreased (Supplemental Figure S1). These informations have been added into Line 94-97, page 2.

-line 44 hydrolytic enzymes

Response: Thanks. The description has been changed to be “hydrolytic enzymes” in Line 41.

-line 163 counted not counte

Response: Thanks. The description has been changed to be “counted”.

-the authors used inflorescence tissue for both transcriptomic and metabolomic analyses, though easily available inflorescence is a complex tissue with many tissue types and cell types of which only the pollen was of interest, the authors should address this issue in the discussion

Response: Thanks for this comment. In fact, in this study, the transcriptomic data are derived from stage 4-7 anthers, and only the metabolomic data are achieved based on inflorescences.

Because the metabonomic detection requires a large number of biological samples and 6 biological replicates are required for each genotype, the small size of Arabidopsis anther could not meet such amount requirements. Therefore, we performed metabolomics analysis with easily available inflorescence tissue in this study, although the inflorescence is a complex tissue with many tissue types and cell types. In order to uncover more anther related data from the metabolome data, we jointly analyzed the metabolome data and the stage 4-7 anther transcriptome data. These description have been added into Line 368-374 of the Discussion section.

-line 582 why 50 cycles in the real time PCR? isn't that too high?

Response: It was a typing mistake. The correct number is 35, and it has been revised to “35 cycles” in Line 557, P14 of the revised version.

-in material and methods the authors should provide more information on the clonings and the protein expression methods used

Response: More information on the clonings and the protein expression methods used have been added to the material and methods section, line 603-609 in page 15-16.

-though the authors state that there are no information about the CSLB03 gene, the sequence analysis did not provide any clues about its function? its homology with other genes?

Response: The Arabidopsis CSL genes contain nine distinct groups: the CSLA, B, C, D, E, F, G, H and J subfamilies [99]. In Arabidopsis thaliana, several members of CSLDs have been demonstrated (references 80, 87, 88, 100). However, the functions of CSLB members have not been reported yet. It was known that the CSLB subfamily contains six family members, and CSLB04 is the closest CSLB family member to CSLB03 according to the phylogenetic analysis result (Supplemental Figure S7), which is consistent with previous reports [99]. The phylogenetic tree has been added as Supplemental Figure S7, and related description has been added into line434-461 of the discussion section. Moreover, the method for constructing the phylogenetic tree of the CSLB subfamily has been added into line 621-626 of page 16.

Round 2

Reviewer 1 Report

According the response from the authors, it is recommended to accept this manuscript.